# Applications of Extracellular Vesicles in Nervous System Disorders: An Overview of Recent Advances

**DOI:** 10.3390/bioengineering10010051

**Published:** 2022-12-30

**Authors:** Safir Ullah Khan, Muhammad Imran Khan, Munir Ullah Khan, Noor Muhammad Khan, Simona Bungau, Syed Shams ul Hassan

**Affiliations:** 1Hefei National Laboratory for Physical Sciences at the Microscale, School of Life Sciences, University of Science and Technology of China, Hefei 230027, China; 2School of Life Sciences and Medicine, University of Science and Technology of China, Hefei 230027, China; 3MOE Key Laboratory of Macromolecular Synthesis and Functionalization, International Research Center for X Polymers, Department of Polymer Science and Engineering, Zhejiang University, Hangzhou 310027, China; 4School of Tropical Crops, Hainan University, Haikou 570228, China; 5Department of Pharmacy, Faculty of Medicine and Pharmacy, University of Oradea, 410028 Oradea, Romania; 6Shanghai Key Laboratory for Molecular Engineering of Chiral Drugs, School of Pharmacy, Shanghai Jiao Tong University, Shanghai 200240, China; 7Department of Natural Product Chemistry, School of Pharmacy, Shanghai Jiao Tong University, Shanghai 200240, China

**Keywords:** exosomes, central nervous system diseases, engineering modification, vehicles, bioengineering

## Abstract

Diseases affecting the brain and spinal cord fall under the umbrella term “central nervous system disease”. Most medications used to treat or prevent chronic diseases of the central nervous system cannot cross the blood–brain barrier (BBB) and hence cannot reach their intended target. Exosomes facilitate cellular material movement and signal transmission. Exosomes can pass the blood–brain barrier because of their tiny size, high delivery efficiency, minimal immunogenicity, and good biocompatibility. They enter brain endothelial cells via normal endocytosis and reverse endocytosis. Exosome bioengineering may be a method to produce consistent and repeatable isolation for clinical usage. Because of their tiny size, stable composition, non-immunogenicity, non-toxicity, and capacity to carry a wide range of substances, exosomes are indispensable transporters for targeted drug administration. Bioengineering has the potential to improve these aspects of exosomes significantly. Future research into exosome vectors must focus on redesigning the membrane to produce vesicles with targeting abilities to increase exosome targeting. To better understand exosomes and their potential as therapeutic vectors for central nervous system diseases, this article explores their basic biological properties, engineering modifications, and promising applications.

## 1. Introduction

The term “central nervous system (CNS) diseases” refers to a broad variety of illnesses, from mild neurological impairments that can manifest as motor, cognitive, visual, linguistic, or cognitive problems alone or in combination with comas and brain death [1,2]. These days, one in six people around the world show signs of having a central nervous system disorder [3,4]. Central nervous system diseases including multiple sclerosis, Alzheimer’s, and Parkinson’s all have high societal healthcare expenditures [5,6]. Despite extensive research, diagnosing and treating CNS diseases remains a clinical challenge [7]. The blood–brain barrier (BBB) is a specialized microvasculature in the brain that prevents harmful substances from crossing over into the central nervous system (CNS) when it is in a normal state [8,9]. Drug transport to the CNS, however, hampered various factors such as enzymatic breakdown, the blood–brain barrier, limited circulation lifetimes, and low tissue penetration. To effectively treat CNS diseases, it is necessary to overcome the BBB’s sieving out of a large proportion of medicines [10].

Consequently, solving these issues has emerged as one of the greatest obstacles in treating brain diseases [11]. Possible solutions include using nanotechnology for controlled medication delivery, imaging, and gene editing, all of which are aided by biological, physical, and chemical alteration tactics [12,13]. The nanoparticle is an effective tool for maximizing distribution efficiency, heightening therapeutic benefit, and decreasing unintended consequences. Intravenous, intranasal, and local injections, such as those administered intracranially, are all being explored as potential delivery mechanisms for the nanoplatform being developed for the diagnosis and treatment of brain diseases [14]. Because of their ability to transport bioactive molecules like proteins and encoding and non-coding RNAs between cells and organs, extracellular vesicles play a crucial role in cellular signaling [15]. Extracellular vesicles (EVs) [16], AAV vectors [17], erythrocyte membrane-encapsulated nanocarriers [18], cell-based delivery nanocarriers [19], injectable hydrogels [20], immunomodulators [21], and many other drug delivery technologies have been created to bypass the BBB. Surprisingly, EVs outperformed them all.

Recently, exosomes have been identified as the most reliable biomarkers for disease diagnosis and the most efficient drug transporters for treating diseases [22,23,24]. Exosomes are endogenous, have good pharmacokinetics and unique immunological characteristics, and may pass physiological barriers, making them superior to synthetic drug delivery vectors like liposomes and nanoparticles [25]. Additional functionalities can be used upon exosomes and for site-specific medication delivery to surface changes [26]. When taken as a whole, these characteristics of exosomes make them helpful in identifying and treating central nervous system disorders.

Exosomes can deliver bioactive substances through a variety of pathways and safely and efficiently transfer bioactive substances to participate in cell metabolisms, such as tissue repair [27], immune regulation [28], and tumor therapy [29]. Exosomes can carry genetic material, have stable lipid membrane structure, and are widely distributed in body fluids and other essential characteristics of delivery carriers. These have gradually become an important direction of disease research as therapeutic carriers. The complex blood–brain barrier (BBB) exists in the central nervous system, which means the treatment of its diseases has certain limitations [30]. To some extent, the brain is shielded by a semipermeable but highly selective barrier called the blood–brain barrier. However, some therapeutic medications have trouble crossing the blood–brain barrier [31]. Exosomes’ delivery carrier benefits allow them to traverse the blood–brain barrier with their cargo. Consequently, it is imperative to research medicine delivery based on exosomes [32,33]. The efficiency with which exosomes transport nucleic acids, enzymes, and small molecule medications can be enhanced by studying their absorption mechanism in cells.

Recent research has shown that exosomes can be helpful in both the diagnosis and treatment of central nervous system disorders. Examples of uses for detecting synuclein in blood exosomes include aiding in differentiating PD and MSA and identifying potential therapeutic targets [34]. GAP43, grain, SNAP25, and synaptotagmin 1 are neuro exosomal synaptic proteins in the blood that are valuable biomarkers for detecting AD 5–7 years earlier cognitive problems [35]. Stroke and nerve injuries can be treated with isolated exosomes alone [36]. Exosomes have been the focus of intensive study due to their potential as biomarkers and therapies for neurological diseases. This article thoroughly overviews the most recent research on exosomes and several severe neurological diseases. We also discuss how exosomes could be used to deliver medications to specific brain parts and treat neurological problems.

## 2. Origin and Biogenesis of Exosomes

The biosynthesis of exosomes begins with the invagination of the plasma membrane (PM), which forms the early endosome and then gradually matures into the late endosome. At later stages, endosomal membranes germinate inward to produce multivesicular bodies (MVBS) and intraluminal vesicles (ILVs). In this process, nucleic acids and proteins are encapsulated in exosomes [37,38]. When MVB fuses with PM, ILVs are free outside the cell, namely exosomes [39,40] (Figure 1). The primary elements of exosomes include lipids, proteins (including mRNA, miRNA, lncRNA, and circRNA), and RNA (including enzymes, heat shock proteins, transcription factors, and tetraglycoproteins) (Figure 1). There are several databases dedicated to exosomes, such as Vesiclepedia, ExoCarta, and exoRBase. Databases like this provide much knowledge about exosomal proteins, lipids, and RNA. However, whereas all exosomes share some exosome components, others are present only in those released by certain types of cells [41]. Exosomes also affect the transcription process of recipient cells by carrying different types of nucleic acids and participate in a regulatory role in cell-to-cell signal communication, organ development, and physiological function [42,43]. Exosomes have many proteins and nucleic acids, laying the foundation for their engineering transformation.

Figure 1 illustrates exosome biogenesis. Endosomes and MVB contribute to the formation of exosomes, which are ILVs; four transmembrane proteins and the ESCRT complex play a role in the maturation of exosomes. When MVB fuses to the plasma membrane, it triggers the release of ILVs, also known as “exosomes”, which are mediated by various Rab proteins (Published in open access Creative Commons Attribution-NonCommercial-NoDerivatives 4.0 International License [44]).

## 3. Subtypes and Classification of Extracellular Vesicles (Evs)

The recommendations made at MISEV2018 allow the use of the generic term “EV” and describe physical characteristics (such as size and density), biochemical elements (such as the composition of surface proteins and lipids), and cell state/origin cells. Although there is still no agreement on how to distinguish the physical or biochemical characteristics of different subtypes, this is the case, making accurate EV classification extremely challenging (e.g., podocyte EV vs. hypoxic EV) [45]. In situations where the identities of EVs cannot be verified with perfect certainty, the term “extracellular particle” is recommended to prevent misunderstanding. The three main types of released vesicles that make up EVs are exosomes, which are produced by endosomes and have a size range of 30 to 100 nm; plasma membrane-derived “microparticles,” also known as ectosomes, microvesicles, or shedding vesicles and apoptotic bodies, which are produced by dying cells and have a size range of 200 to 5000 nm [46,47]. Although exosomes are created when the plasma membrane merges with multi-vesicular bodies (MVBs) that develop from the invagination of the late endosomal membrane to form intraluminal vesicles, microparticles are produced by the plasma membrane bursting outward (ILVs). However, the astounding range in EV size, form, and freight challenge this traditional way of categorization.

Large oncosomes, which can be anywhere from one to ten micrometers in length, are secreted by malignant cells and carry oncogenic material [48,49]. Exophers, which are around four micrometers in length, contain protein clusters and organelles. Migrasomes, which are greater than one micrometer in length, are secreted during migracytosis, which is a cell migration-dependent process of vesicular secretion. Within EV isolates, all three atypically big EV subtypes have been seen. The detection of exomeres, which are smaller EV isolates, has also been made possible by the application of asymmetric flow field-flow fractionation [50]. Exomeres are tiny, non-vesicular populations that are morphologically, physiologically, and biochemically distinct from exosomes (50 nm or less in size). Although the details of their composition are still unknown, the finding demonstrates that they are not vesicles with a lipid membrane, but rather a collection of molecules with a different protein, N-glycosylation, lipids, RNA, and DNA profiles. Exomeres have different biodistribution patterns than exosomes, and they are enriched in proteins that are important for metabolism. Supermeres are a group of extracellular nanoparticles that were only recently found in the exomere supernatant. Unlike other exomere-associated nanoparticles, these supermeres differ from them in terms of morphology and chemical structure. Supermeres include a large number of enzyme complexes, transport more extracellular RNA, and also have a high rate of cellular absorption compared to small-EVs and exomeres [51].

## 4. Exosomes’ Neuroprotective Properties

These vesicle-producing cells include exosomes. Exosome production and composition provide clues about the viability of the cells they came from. Exosomes also give cells a way to change the ECM’s molecular makeup and functionality [52,53]. Additionally, exosomes communicate molecules and signals across intercellular vesicle traffic routes, causing paracrine or distal CNS effects that are local. Exosome makeup and cellular state-mediated reactions can affect how a disease develops or is suppressed.

Additionally, modified exosomes can deliver a variety of therapeutic loads, such as immune modulators, antisense oligonucleotides, chemotherapeutic medicines, and small interfering RNAs [54]. In this study, we focused on specific biological effects of exosomes associated with CNS disorders, although these effects cover many physical aspects. The exosomes secreted by neurons, glial cells, and other CNS cells help to create the intricate web of interrelated data that serves as the system’s physiological and pathological foundation [55]. Exosomes’ protective actions in CNS disorders include, among others [56], the promotion of angiogenesis, regulation of immunity, inhibition of neuronal death, and promotion of myelination and axonal development.

### Improving Angiogenic Potential

Endothelial cells, neurons, adipose tissue, and immune cells are just a few of the cell types from which exosomes are derived, all of which stimulate angiogenesis in their unique ways (Table 1). Exosomes produced from endothelial cells increased angiogenesis by secreting miR-214, downregulated ataxia telangiectasia mutant (ATM) expression in other endothelial cells [57]. Exosomes secreted by adipose-derived mesenchymal stem cells (MSCs) stimulate angiogenesis through two mechanisms: by suppressing DLL4 (Delta-like ligand4) expression and by increasing apical cell formation [58]. Mesenchymal stem cells (MSCs) isolated from human umbilical cords can trigger the Wnt4/-catenin pathway, which in turn stimulates blood vessel growth [59]. Bone marrow mesenchymal stem cells (BMSCs) secrete exosomes that start the nuclear factor-kappa B (NF-kB) course and transport the transcription factor pSTAT3 [60,61]. Exosomes secreted by human placental-derived mesenchymal stem cells (MSCs) induced angiogenesis-related gene expression and promoted endothelial tube production and migration [62]. Furthermore, an in vivo investigation showed that infusion of PlaMSC-Exo might enhance angiogenesis in a mouse auricle ischemia injury model [63]. Exosomes isolated from human UCB exhibited miR-21-3p, stimulating fibroblast proliferation and migration and boosting endothelial angiogenic activity in in vitro and in vivo investigations [64].

Although exosome contents vary greatly depending on their point of origin, they always share a plasma membrane-derived lipid bilayer and collection of endocytosis amino acids and nucleic acids (DNA, RNA) (published in open access Creative Commons Attribution-NonCommercial-NoDerivatives 4.0 International License [68]).

## 5. Carrier Characteristics and Engineering Modification Mechanism of Exosomes

Engineered exosomes include exosomes loaded with specific contents or surface modified [69]. Natural exosomes have a low capacity to target, but manufactured exosomes have a far better targeting ability [26]. Exosomes are characterized by their high transport efficiency, low immunogenicity, strong biocompatibility, and capability of readily crossing the blood–brain barrier. Based on exosomes’ natural lipid bilayer structure, exosomes are engineered using specific surface molecules. This engineering strategy not only preserves the intrinsic characteristics of exosomes, but more importantly, the engineered exosomes can also effectively coat therapeutic drugs (Figure 2). Including loading miRNA [70], curcumin (Cur) [71], and dopamine [72] and then targeting the big brain delivery.

Genetically engineered dendritic cells express the lysosomal-associated membrane glycoprotein 2b (Lamp2b). The RVG peptide, targeted by the rabies virus glycoprotein (RVG), causes vulnerable cells created through engineering to release exosomes containing the RVG peptide [68,73] (FIG. 2). This RVG peptide can selectively target neurons and brain endothelial cells by binding to the nicotinic acetylcholine receptor (nAChR) [74]. Notably, exosomes modified by RVG peptide showed the targeting ability to cross the blood–brain barrier when injected into mice. These findings will lay the foundation for studying exosomes as delivery vehicles across the BBB based on nucleic acid and protein therapies.

Exosome-producing cells were transfected with the pcDNA GNSTM-3- RVG-10-Lamp2b-HA vector [75], which led to the creation of engineered cells with RVG and GNSTM peptides on their surfaces and the collection of these designed exosomes. Then, via continuous extrusion, exosomes were created that were coated in gold nanoparticles (AuNPs) [76]. According to the findings, surface-modified exosomes on AuNPs demonstrated more BBB penetration than total exosomes [77]. This research could hasten the creation of generic design guidelines for AuNP surface changes across the BBB and offer brand-new treatment options for neurological disorders.

## 6. CNS Cell-Release Exosomes and Their Potentially Bidirectional Pathway

Exosomes may promote neurodegeneration and neuroinflammatory processes in the central nervous system (CNS). But they might also be active protective components crucial for average CNS growth and function [78,79]. Increased quantities of microglial exosomes have been seen in AD patients, and exosomes produced by oligodendroglioma cells are neurotoxic [80]. Exosomes are released by various cell types, including those in the nervous system (Figure 3), including neurons, microglia, neural stem cells, hepatocytes, and neuro embryonic stem cells [81,82]. It makes sense considering exosome function in both normal and pathological central nervous system states. It is intriguing to consider the possibility that these exosomes can cross the BBB, etc., and the blood barrier and go from the CNS to the peripheral to the SNC in both normal and pathological conditions.

Exosomes are small vesicles that can travel through the bloodstream and trigger the malignant transformation of normal cells [83]. Malignant glioma cells generate and release exosomes into circulation, specifically initial tumors originating from the glial stem or progenitor cells. Exosomes originating from glioblastoma (GBM) have been isolated from patient blood, and their mRNA profiles have been compared to those of the primary tumor cells using the immunomagnetic exosome RNA (iMER) analysis platform [84]. Exosomes transport misfolded proteins or proinflammatory chemicals as they circulate in the blood and cerebral fluid in a variety of neurodegenerative diseases, including frontotemporal dementia (FTD), amyotrophic lateral sclerosis (ALS), temporal atrophic lateral sclerosis (FTD-ALS), Parkinson’s disease (PD), bovine dermopathy, and Alzheimer’s disease (AD) [85,86,87]. The BBB can be bypassed by using a modified form of the rabies virus glycoprotein (RVG) to direct EVs to carry siRNA over the blood–brain barrier [88]. Blood-borne macrophages were genetically modified to transport nano-enzymes made from antioxidant proteins [89]. Therapeutic proteins have been proven to traverse the BBB, and exosome release is hypothesized to be one way macrophages deliver nano-masses to their intended recipient cells [90]. Furthermore, it has been shown that EV absorption by neurons in vitro (PC12 cells, a neuronal rat adrenal pheochromocytoma cell line) and neurons and microglia in mice is more efficient than other traditional vector liposomes [91].

**Figure 3 bioengineering-10-00051-f003:**
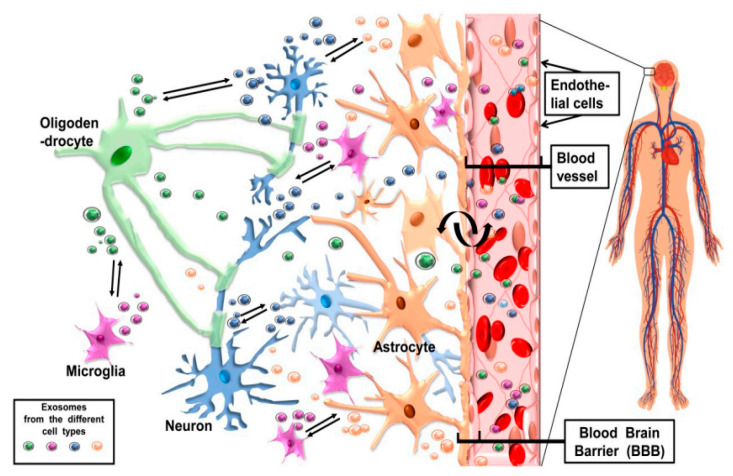
Brain and exosome flow diagram shows cells in nerve tissue and blood arteries. Shown are blood vessel epithelial cells, the blood–brain barrier (BBB), and nerve cell exosomes. Published under the term of common creative license [92].

## 7. Application of Exosome Vector Therapy in Central Nervous System Diseases

### 7.1. Exosomes as Drug Carriers

Stroke is a frequent neurological disorder that can affect any part of the brain or spinal cord [93]. Neuronal apoptosis, oxidative stress, and the inflammatory response are crucial pathogenic processes of ischemic brain injury [94]. Numerous biological actions of curcumin include anti-inflammation, anti-oxidative stress, and anti-apoptosis [95]. However, its clinical applicability is restricted because of its weak water solubility and unstable metabolism in vivo. Exosomes can be wrapped in curcumin for brain delivery, improving ischemic brain injury [96]. Scientists took macrophages as donor cells, co-incubated them with 3 μg/mL curcumin at 37 °C for 24 h, and obtained macrophage exosomes loaded with curcumin (Exo-Cur) by hypercentrifugation [97]. ExoCur can significantly reduce the accumulation of reactive oxygen species (ROS) and maintain the integrity of BBB by activating the expression of tight junction proteins such as occludin and claudin-5.

Scientists conjugated the functional ligand RGDyk cyclic peptide to the surface of exosomes through the bioorthogonal reaction between the exosomes modified with cyclooctene and azide polypeptides [98]. After being loaded with curcumin, the functional ligand RGdyK cyclic peptide could be targeted to the ischemic site of the stroke mouse model, effectively inhibiting the inflammatory response and cell apoptosis in the lesion area [99]. Researchers took mouse embryonic stem cells (mESC) as donor cells and incubated them with curcumin for 15 min at room temperature, followed by rapid freeze-thaw cycles of two to three times, and obtained mESC exosomes loaded with curcumin by ultra-centrifugation [100]. Compared with the control group, exosomes loaded with curcumin significantly increased the neurological function score, reduced the infarct volume, improved the inflammatory response, and reduced astrogliosis and N-methyl-D-aspartate receptor 1(N-methyl-D-aspartate receptor 1) in the stroke model group. Expression of NMDAR1. In addition, exosomes loaded with curcumin can increase drug solubility, stability, and bioavailability and inhibit lipopolysaccharide (LPS)-induced brain inflammation caused by microglial activation [101]. After intranasal administration, microglia can also take up exosomes loaded with gingerol, thereby reducing LPS-induced brain inflammation [102]. The experimental results of drug-loaded exosomes and intranasal delivery routes offer hope for potential non-invasive treatment of stroke in the future.

The most prevalent neurodegenerative disease is Alzheimer’s (AD), distinguished by the steady buildup of phosphorylated Tau protein [103]. According to one study], curcumin-loaded exosomes from macrophages can prevent Tau phosphorylation by OA through the Akt/GSK-3 signaling pathway, enhancing cognitive performance in AD mice [104]. This work suggests that lymphocyte function-related antigen 1 (LFA-1) may also be inherited by exosomes generated from macrophages. This protein facilitates exosome movement across the BBB and improves curcumin’s capacity to traverse the BBB by interacting with intercellular adhesion molecule-1 (ICAM-1). Quercetin (Que)-loaded plasma exosomes (Exo-Que) increase brain Que targeting and medication bioavailability [105]. They also discovered that Exo-Que prevented Tau cyclin-dependent kinase 5 (CDK5)-mediated phosphorylation, reducing nerve fiber entanglement to enhance the mice used as an AD model’s cognitive performance [105,106].

To treat brain tumors, exosomes can also be coated with paclitaxel and doxorubicin [107,108]. Glioblastoma (GBM)-derived exosomes loaded with doxorubicin were incubated with GBM at 37 °C for 4 h. Compared with doxorubicin alone, the treatment of doxorubicin-loaded exosomes inhibited the proliferation of GBM more effectively [109]. This also indicates that GBM exosomes may have a homing effect and can be used as an ideal carrier for delivering therapeutic drugs to propriocellular cells [110].

### 7.2. Exosomes as Protein Carriers

Exosomes can also be loaded with exogenous proteins for brain delivery, thereby improving neuronal apoptosis after stroke. Pigment epithelial-derived factor (PEDF) is a protein with many uses, including its ability to reduce inflammation, neutralize free radicals, and protect nerve cells [111,112]. Researchers transfected adipose-derived mesenchymal stem cells (ADSCs) with a PEDF overexpression vector and then obtained exosomes containing PEDF with an exosome extraction kit [113]. After intraventricular injection, It was found that PEDF carried by exosomes could regulate the expression level of autophagy-related proteins and significantly improve the apoptosis of cerebral ischemic nerve cells. It was mixed macrophage exosomes with brain-derived neurotrophic factor (BDNF) in 10 mmol/L phosphate buffer, loaded BDNF as a model protein into exosomes, and after intravenous administration [90]. It has the ability to transport BDNF across the blood–brain barrier, where it can reduce neuroinflammation. Researchers discovered that inflammation in the brain amplified this effect of transmission. This study has significant ramifications for using macrophage exosomes as nanocarriers to transport therapeutic proteins to the brain to treat CNS disorders.

Macrophages were transfected with plasmid DNA encoding catalase and obtained exosomes containing substances required for catalase synthesis (including DNA, mRNA, transcription factors, or catalase protein) [114]. These contents can be efficiently transferred to neurons by exosomes and play a neuroprotective role in Parkinson’s disease (PD) model mice. This study demonstrates that exosomes are an efficient delivery system for proteins and genetic material to target cells. Later, mouse macrophages were used as donor cells to isolate and purify exosomes by hypercentrifugation and evaluated four methods of exosome loading catalase: incubation at room temperature, freeze-thaw cycle, ultrasonic treatment, and continuous extrusion [115]. The catalase-loaded exosomes were then separated by gel filtration chromatography. Intranasal administration of these exosomes loaded with catalase significantly reduced the proliferation of microglia and astrocytes in the mouse brain and promoted the survival of neurons in the PD model mice. This study shows that exosomes are effective carriers of therapeutic protein catalase. Catalase is encapsulated in exosomes, which can retain its bioactivity for a long time, prolong its blood circulation time, and reduce its immunogenicity, thus improving the therapeutic effect.

### 7.3. Exosomes as Nucleic acid Carriers

In central nervous system diseases, using exogenous exosomes such as stem cell-derived exosomes, carrying genetic material, especially microRNA, can improve the apoptosis process of neural cells after stroke. Mesenchymal stromal cells (MSC-EXOs) transport miR-17-92, which promotes neurogenesis and reduces neuronal death by activating the PI3K/protein kinase B/rapamycin/glycogen synthase kinase 3 signaling pathway [116,117]. miR-29b-3p carried by MSC-EXO can also improve angiogenesis and neuronal apoptosis in stroke model rats by activating the PTEN-mediated Akt signaling pathway [118]. It was found that exosomes fused with RVG peptide and Lamp2b could effectively deliver miR-124 to the ischemic brain site and promote neurogenesis in ischemic brain tissue [119]. This study suggests that work exosomes may be the best candidate for transporting gene drugs to treat stroke. it was found that the engineered exosomes as delivery vectors could prevent miRNA degradation and improve stroke symptoms [120]. The team loaded the exosomes modified by RGDyk cyclic peptide with miR-210 and injected them through the tail vein. They could target the ischemic brain tissue of mice. Furthermore, the expression of vascular endothelial growth factor (VEGF) and angiogenesis in the lesion area was enhanced. In addition, exosomes modified by miR-133b can significantly inhibit the expression of RhoA and activate the ERK1/2 -CREB signaling pathway after intracerebral hemorrhage (ICH) stroke, thus playing a neuroprotective role [121].

Exosomes released from microglia can transport miR-146a-5p to neurons, where it can then downregulate its synaptic targets and alleviate neuroinflammation [122,123]. Exosomes modified with RVG peptide were injected through the tail vein to deliver miR-193b-3p into the brains of subarachnoid hemorrhage (SAH) mice [124]. It can inhibit the expression of histone deacetylase 3 (HDAC3) after SAH and promote the acetylation of NF-κB p65 to reduce neuroinflammation. This study provides a new strategy for delivering miRNA from exosomes to treat SAH.

In the treatment of neurodegenerative diseases, injected exosomes carrying miR-29b into the hippocampus of AD rat models, which could improve AD animals’ spatial learning and memory ability by reducing the expression levels of target genes NAV3 and BIM [125]. By surface electroporation, BACE1 siRNA was loaded into exosomes, and then intravenously administered to AD model mice, where it suppressed BACE1 mRNA and protein expression and, in turn, reduced amyloid-beta (A) peptide synthesis in AD mice. As a protective measure, RVG peptide-modified exosomes can reduce alpha-synuclein (-Syn) mRNA and protein levels in PD model mice by delivering siRNA to brain tissue [126]. In addition, exosomes loaded with short hairpin RNA micro circle (shRNA-MC) can deliver the contents to the brain tissue of PD model mice after injection through the tail vein. Furthermore, α-Syn aggregation level and dopaminergic neuron loss are reduced [127].

MiR-133b-carrying MSC-EXO have been shown to suppress the promoter of the Zeste 2 (EZH2) gene, consequently reducing GBM proliferation, penetration, and migration during tumor therapy [128]. ArfGAP’s GTPase domain, ankyrin repeat, and PH domain 2 can all be repressed by miR-199a that is transported in exosomes (ArfGAP with GTPase domain, ankyrin repeat, and PH domain 2) control of glioma by AGAP2 [129]. The payload also contains miR-146b-5p. Glial cells are susceptible to having their expression of the epidermal growth factor receptor (EGFR) suppressed by exosome 1655 because of the RNA binding ability of this particle. Additionally, glioma malignancy and aggressiveness can be lessened [130]. Some studies also found that the expression of miR-1 in exosomes is reduced during the growth and invasion of GBM, which promotes the growth and invasion of GBM [131]. Therefore, further research based on microRNA carried by exosomes may help find new strategies for treating brain tumors.

## 8. Exosome in the Diagnosis and Treatment of Different CNS Diseases

As research into exosomes has progressed, exosomal drug carriers have been the subject of a number of preclinical research on animal models of CNS diseases to determine their efficacy and pave the way for their eventual clinical translation.

### 8.1. Alzheimer’s Disease

Alzheimer’s disease (AD) is a neurological ailment that predominantly manifests itself in the form of cognitive dysfunction and memory loss [132,133]. Drugs now on the market relieve symptoms, but there are no cures. Alzheimer’s disease (AD) pathology is characterized by neuronal cell death, loss of synapses, neurotoxicity, and brain damage. Neurogenic fibrillary tangles (NFT), which are made up of amyloid (A) plaques and improperly phosphorylated tau protein (p-tau), are another characteristic of AD pathology [134,135]. Since exosomes are packed with proteins, RNAs, and other biomolecules, they can serve as a go-between for messages sent between cells [136]. This means that exosomes are highly effective RNA carriers [137,138]. Many RNA medicines are in common use, and one of their effects is gene silencing [139,140]. One example is small interfering RNAs (siRNAs). Technology for delivering siRNAs has improved quickly in recent years, although extrahepatic and, in particular, brain-targeted delivery are still in the testing phase. The first attempt at employing exosomes as vehicles for siRNA delivery for intracerebral administration was carried out by the authors of [73,141]. In order to reduce immunogenicity, exosomes produced from the study participants’ own dendritic cells were modified with RVG neuro-targeting peptides. Therapeutic target BACE1 in Alzheimer’s disease (AD) was successfully knocked down in mice by carefully chosen small interfering RNAs (siRNAs).

### 8.2. Huntington’s Disease 

Many members of the same family can be affected by the neurological condition known as Huntington’s disease (HD) [142]. Neurons begin to die as a result of abnormal Huntington’s proteins that are produced abnormally by Huntington’s genes, which are also known as disease-causing genes [143]. Therefore, silencing this gene could be used as a treatment for HD. When siRNA was treated with exosomes, a common method for siRNA transport sterically changed the siRNA to boost its loading efficacy [144]. If exosomes containing siRNA were injected unilaterally into the striatum of mice, the siRNA would diffuse to the opposite striatum. They brought it to the brain’s neurons, where it caused a dose-dependent reduction in Huntingtin mRNA and protein.

### 8.3. Brain Tumor

About 40% to 50% of all primary intracranial tumors are gliomas, making them the most prevalent type of brain tumor. The prognosis is dismal and there is a significant mortality rate for gliomas [145]. The inability of currently available medications to cross the BBB and build significant local concentrations of drugs at the tumor site is a major factor in the failure to achieve the desired therapeutic effect. Glioma treatment has benefited greatly from exosomes due to their capability to transport drugs across the blood–brain barrier. Recent research has modified exosomes with peptide transactivators and Angiopep-2, for instance. By transporting doxorubicin over the blood–brain barrier, it not only doubled the survival rate of mice with glioma but also considerably mitigated the drug’s harmful effects [146]. Electroporation was used to load exosomes with both SPIONs and curcumin (MRI contrast agents). Finally, they used click chemistry to add peptides that specifically target gliomas to the exosomes, demonstrating a synergistic effect against the tumors. 

### 8.4. Parkinson’s Disease

Parkinson’s disease, often known as PD, is a neurodegenerative disorder that affects a sizable fraction of the global population. PD is characterized by the degenerative loss of dopaminergic (DA) neurons [147,148]. It has been demonstrated that catalase protects neurons. This leads to investigating several strategies for transporting catalase via exosomes from RAW264.7 cells. Exosomes containing catalase were used by neurons in the brain and microglia of PD animals after being administered intranasally, and they showed strong neuroprotective benefits. Similar results were seen when it was transfected HEK-293T cells with a plasmid carrying both peroxidase mRNA and RVG [149]. Exosomes containing peroxidase mRNA were created in vivo by the cells, injected subcutaneously, and transported to the brain, where they reduced neurotoxicity and neuroinflammation induced by PD. Besides the aforementioned methods, dopamine is also utilized to treat PD; however, there are difficulties with dopamine reaching the brain. To alleviate PD mouse symptoms, dopamine was encapsulated in blood exosomes and successfully injected intravenously into the brain regions striatum and substantia nigra.

### 8.5. Stroke

Both ischemic and hemorrhagic stroke, which include blood artery rupture or occlusion, are collectively referred to as stroke [150,151]. An estimated 152 million people worldwide are living with this neurological disorder [152]. A significant contributing factor to hemorrhagic stroke is subarachnoid hemorrhage. The inflammatory response was reduced by inhibiting HDAC3 with the use of exosomes that were injected intravenously into mice brains [124]. Last but not least, neurobehavioral abnormalities, cerebral edema, BBB damage, and neurodegeneration were all improved.

Ischemic stroke, which is more common than its hemorrhagic counterpart [153], results from a constriction or closure of a cerebral artery. Many microRNAs (miRNAs) have been shown to have therapeutic promise in a range of central nervous system (CNS) diseases [154], and miRNAs themselves play a role in regulating CNS development, evidenced by brain restoration and suppression of neuroinflammation [155], MSC-exosomes have proven effective in the treatment of a wide range of neurological diseases [156,157]. As a result, MSC-exosome-encapsulated miRNAs have been investigated as potential therapeutic agents for treating ischemic strokes in a number of studies [116,158].

Curcumin is a polyphenolic molecule found in nature that has been shown to have strong anti-inflammatory effects [159]. Co-incubation allowed curcumin to penetrate the exosomes that had been linked to RGD peptides by bioorthogonal chemistry. Following intravenous administration, drug-loaded exosomes localized to the brain, where it inhibited the inflammatory process and apoptosis in regions affected by ischemic brain damage. Exosomes from rat macrophage RAW264.7 were loaded with curcumin to take advantage of their targeted migratory potential, as opposed to effector brain targeting of exosomes. When the exosomes were administered intravenously, they made their way to the brain, where they shielded neurons from harm by limiting the release of reactive oxygen species (ROS) by protecting the blood–brain barrier (BBB). The exosomes released by mouse ESCs have been shown to bind curcumin, suggesting that this compound could be used to treat stroke.

Coating exosomes with active chemicals like curcumin and miRNAs could make them more effective therapeutic tools for treating strokes. Other active substances such as circSCMH1, pigment epithelium-derived factor (PEDF), and recombinant human NGF mRNA are also potential for coating with exosomes [113].

### 8.6. Alternate Disorders of the Brain: Viruses and Drug Addiction Issue

When the drug ceases activating the central nervous system, a cascade of withdrawal symptoms begins that can have devastating effects on both the addict and society as a whole. However, the number of therapies is extremely constrained [160]. Opioids, for example, have their analgesic and addictive actions mediated via MOR receptors.

Multiple methods allow viruses to enter an organism and spread throughout the host cells. The ZIKA virus, for example, can infect the central nervous system by crossing the placenta and the BBB to reach the developing brain of the fetus, where it can cause microcephaly. When injected into the mother’s bloodstream, the RVG-modified exosome generated from HEK 293T cells expressing ZIKV-specific siRNA was able to reach the fetal mice, penetrate the placental barrier, and target the brain, protecting the animals against ZIKV infection and reducing neuroinflammation.

In conclusion, exosomes have seen extensive application as central nervous system medication delivery agents. They are capable of carrying a variety of medicinal chemicals, such as proteins, nucleic acids, and tiny molecule substances. The treatment of neurodegenerative disorders, neuroinflammatory disorders, brain malignancies, and brain viral infections have all shown promising outcomes when using exosomal drug delivery methods. This is highly encouraging for further investigation and application of exosomes as drug-delivery agents for the treatment of CNS disorders. Table 2 summarizes the research on exosomal medication delivery methods for CNS disorders.

## 9. Conclusions

The most recent database (http://www.exocarta.org, accessed on 23 September 2022) shows that exosomes contain 9769 proteins, 2838 microRNAs, and 3408 mRNAs. Given their voluminous contents, exosomes are increasingly being recognized for their crucial roles in cellular communication, disease pathogenesis, and therapeutic intervention. It sheds light on the potential of exosomes as medication delivery mechanisms in the central nervous system.

As an alternative to synthetic nanoparticles, exosomes have several advantages when it comes to delivering drugs to the brain. These qualities include being able to pass the blood–brain barrier, being non-toxic, having a large payload-carrying capacity, being protective, and having low immunogenicity [101,168,169]. Exosome surface engineering has frequently sought to improve the efficacy of exosome entry into the central nervous system. It can improve therapeutic efficacy by increasing local medication concentration at the lesion site while decreasing harmful side effects. Nonetheless, the method by which modified exosomes improve the efficiency with which they target the central nervous system and the stability of their structure and function are still largely unknown. More study is required to improve the reliability, safety, and uniformity of such approaches.

Nucleic acids (mRNAs, miRNAs, shRNAs, siRNAs, etc.), small molecule medicines (dopamine, doxorubicin), natural components (paclitaxel, curcumin, resveratrol), and particular proteins can all be found in exosomes (pigment epithelium-derived factor, catalase). There are a variety of pre- and post-exosome-secretion loading strategies for therapeutic medicines and exosomes. There have been experiments using electroporation, extrusion, sonication, electroporation with co-incubation, genetic editing, and repeated freezing and thawing. Therefore, it is important to learn more about the processes involved in exosome internalization so that we can choose the most effective drug-loading strategies for various therapeutic compounds. Exosomes that contain the aforementioned therapeutic substances have shown promise in the treatment of a number of central nervous system (CNS) disorders, including cancer, multiple sclerosis, degenerative diseases, brain or spinal cord injuries, drug addiction, strokes, and viral infections. However, more mechanistic research is needed, and improvements can be made to exosome extraction and purification before exosomes can fully live up to their potential as a CNS drug delivery method.

The practical application of exosomes as a mechanism for drug transport to the central nervous system is influenced by multiple factors, including adjustments to the engineering, route of administration, type of therapeutic molecule, and encapsulation mode. More research into the current problem, a better understanding of the unknown mechanism, and the full use of its clinical potential will help the vast majority of patients with CNS disease. 

## Figures and Tables

**Figure 1 bioengineering-10-00051-f001:**
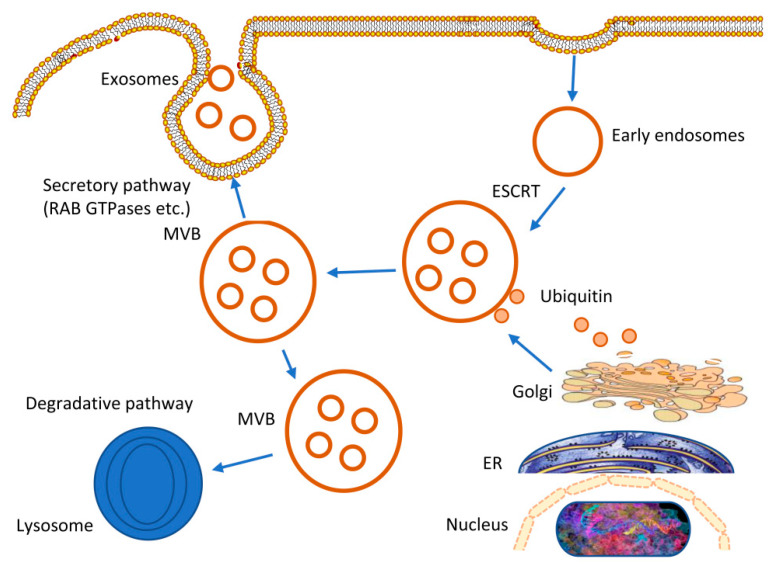
Exosome structure, biogenesis, and secretion.

**Figure 2 bioengineering-10-00051-f002:**
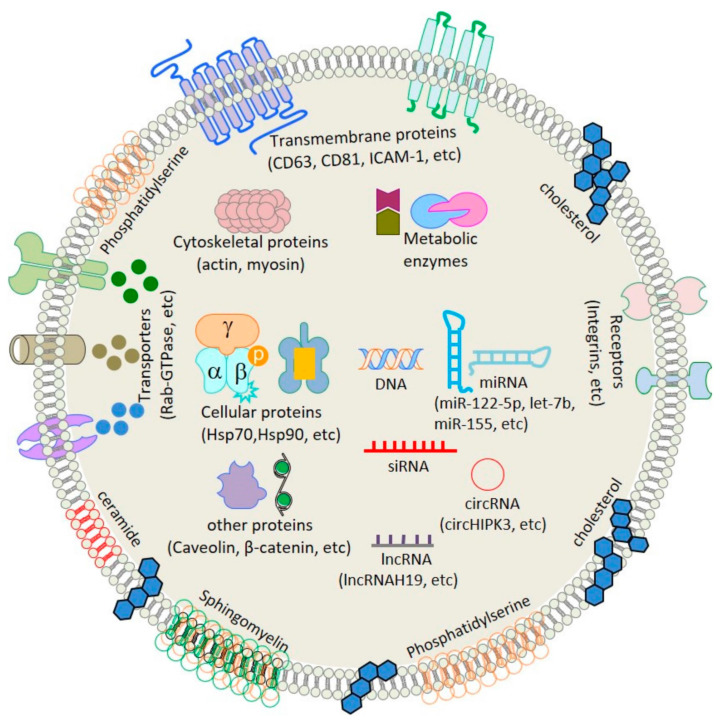
General representation of the bioengineered exosome structure.

**Table 1 bioengineering-10-00051-t001:** Exosome-induced angiogenesis.

Exosome Origin	Transition	Downstream Route or Molecule	Cells or Models	Downstream	References
	molecule			effect	
Endothelial cell	miR-214	ATM	Endothelial cells	−	[57]
Adipose-derived MSCs	miR-125a	DLL4	Endothelial cells	−	[58]
MSC from umbilical cord	NO	Wnt4/β-catenin	NAO	+	[59]
Bone-marrow MSCs	pSTAT3	NF-κB	NO	+	[60]
Human term PlaMSCs	NA	Angiogenesis-related gene	In vivo murine auricle	+	[61]
			Ischemic injury model		
Human UCB	miR-21-3p	NO	Fibroblasts, endothelial cells	NO	[64]
Induced vascular progenitor cells	NO	NO	Rat hindlimb ischemia model	NO	[65]
Human EPC	NO	NO	H/R induction	NO	[66]
Hypoxic-induced MSC	NO	NO	NO	NO	[67]

**Table 2 bioengineering-10-00051-t002:** Preclinical research on the use of exosomes as DDS in the treatment of CNS disorders.

Disease	Therapeutic Molecule	Donor Cell	Modification Strategy	Drug Loading Method	Administration Route	Animal	Targeted Cells	Ref.
Alzheimer’s disease	BACE1 siRNA	self-derived dendritic cells	Lamp2b-RVG	electroporation	intravenous	Mice	neurons, microglia	[73]
Parkinson’s disease	a-Syn siRNA	primary dendritic cells	Lamp2b-RVG	electroporation	intravenous	Mice	unknown	[126]
	aptamer F5R1	HEK-293T cells	Lamp2b-RVG	co-incubation	intraperitoneal	Mice	microglia, neurons	[161]
	catalase	RAW264.7	None	sonication orextrusion	intranasal	Mice	neurons andmicroglia	[115]
Huntington’sdisease	hsiRNAHTT	glioblastoma U87cells	co-incubation	None	Unilateral brain infusion	Mice	neurons	[162]
Stroke	curcumin	RAW264.7	None	co-incubation	intravenous	Rats	neurons and endothelium cells	[97]
	curcumin	mouse embryonic stem cells (MESCs)	None	co-incubation	intranasal	Mice	astrocytes and neurons	[100]
	recombinant human NGF mRN	HEK-293T cells	Lamp2b-RVG	transfection	intravenous	Mice	microglia, neurons	[163]
	PEDF	stem cells (ADSCs)	None	transfection	intravenous	Rats	unknown	[113]
	circ SCMH1	HEK-293T cells	Lamp2b-RVG	transfection	intravenous	Mice and rhesusmonkeys	microglia, neurons, and astrocytes	[164]
Brain tumor	doxorubicin	brain endothelial cell (bEND.3)	None	co-incubation	intravenous	Zebrafishes	unknown	[108]
	SPIONs	RAW264.7	RGE-peptide	electroporation	intravenous	Mice	glioma	[165]
ZIKV infection	ZIKV-specific siRNA	HEK-293T cells	Lamp2b-RVG	electroporation	intravenous	Mice	microglia, neurons	[166]
Morphine addiction	Mu (MOR) siRN	HEK-293T cells	Lamp2b-RVG	transfection	intravenous	Mice	neuro2A	[167]

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
