# Peer review of "Applications of Extracellular Vesicles in Nervous System Disorders: An Overview of Recent Advances"

_bioengineering, 2022, doi:10.3390/bioengineering10010051_

Round 1

Reviewer 1 Report

The review “Development in the utilization of bioengineered Exosome vectors for treating diseases of the central nervous system” mentioned about the potential of exosomes to be used in the treatment of several severe neurological diseases. This has great importance at the current scenario as the review on the above context are very less.

However, the review needs major corrections before consideration to be published.

1.      The title needs to be modified (choose a better and appropriate title)

2.      English needs to be improved. Also, the article has typos and incomplete sentences

e.g.

Page 4 “The term "exosome" was coined by by famous scientist to describe vesicles originating in the plasma membrane”

The definition of exosomes mentioned 2 times in the text.

In the text (FIG.1, Figure), in the figure legend it is written as “Figure”. There is no consistency. The authors should strictly follow the journal’s guidelines.

Also, the incomplete sentence also presents at other places that need needs to be thoroughly checked.

3.      The content needs to be re-organized. e.g., The section “Origin and biogenesis of exosomes” needs to be shifted above the section “Exosomes' neuroprotective properties”.

Also, the heading of each section to cross checked and to be made more appropriate.

4.      Table 1. Exosome-induced angiogenesis. Each information in the tables to be supported by appropriate references.

5.      Also, figure legends are not appropriate. e.g. “Figure 2. the Bioengineerined Exosome structure is generally represented.”

6.      There must be separate table for exosome as carrier (drug, protein and nucleic acid carrier) in the treatment of central nervous system diseases with appropriate references.

7.      Conclusion and future perspective sections needs to re written especially future perspectives mentioning the potential as well as challenges and proposal if any.

Author Response

The review “Development in the utilization of bioengineered Exosome vectors for treating diseases of the central nervous system” mentioned about the potential of exosomes to be used in the treatment of several severe neurological diseases. This has great importance at the current scenario as the review on the above context are very less.

Response: Thank you so much to the reviewer for the encouragement and positive aspects of our review, We have responded to all your comments and we hope you will be satisfied by our comments and thanks once again for giving such positive comments for improving our article.

  1. The title needs to be modified

Author’s response: We thank the reviewer for affirmative response on our manuscript. We have gone through the title and modify it accordingly.

  1. English needs to be improved. Also, the article has typos and incomplete sentences e.g. Page 4 “The term "exosome" was coined by by famous scientist to describe vesicles originating in the plasma membrane” The definition of exosomes mentioned 2 times in the text. In the text (FIG.1, Figure), in the figure legend it is written as “Figure”. There is no consistency. The authors should strictly follow the journal’s guidelines. Also, the incomplete sentence also presents at other places that need needs to be thoroughly checked.

Author’s response: We appreciate the reviewer concern and kind reminder, Manuscript has been carefully rechecked to rectify all the spelling mistakes and reference format problems as suggested by the reviewers.

  1. The content needs to be re-organized. e.g., The section “Origin and biogenesis of exosomes” needs to be shifted above the section “Exosomes' neuroprotective properties”. Also, the heading of each section to cross checked and to be made more appropriate.

Authors response: We appreciate the reviewer concern and kind reminder, we have revised the whole manuscript and added  figures based on real scientific data .

  1. Table 1. Exosome-induced angiogenesis. Each information in the tables to be supported by appropriate references.

Author’s response: The authors appreciate the constructive feedback from the reviewer. We have glad to inform that we have added extra for column for references.

  1. Also, figure legends are not appropriate. e.g. “Figure 2. theBioengineerinedExosome structure is generally represented.”

Author’s response: Authors are thankful for this comment. legends of figure 2 has been resolved accordingly .

  1. Conclusion and future perspective sections needs to re written especially future perspectives mentioning the potential as well as challenges and proposal if any

Author’s response: The authors appreciate the constructive feedback from the reviewer. We have glad to inform that we have update the conclusion parts accordingly based on scientific data.

Reviewer 2 Report

Please include proper exosome classification according to International Society for Extracellular Vesicles. It is therefore incorrect to use the names "exosome" and "exosome complex" an ld please indicate the difference.

I appreciate your work in the field of exosomes. However please add section on classification of exosomes to clarify that all publications that are included used the same type of vesicles.The discovery of exosomes was made byconfirming that the transferrin receptors in reticulocytes are associated with small vesicles, which are then secreted from the maturing reticulocytes into the extracellular space . The  term exosome; was clarified a few years later by Rose Mamelak Johnstone, the head of the Department of Biochemistry at the Faculty of Medicine at McGill University in Canada 27 . In 1997, a structure called the “exosome complex” or - often wrongly - the human “exosome”, was additionally discovered. These two concepts should not be confused, as the exosome complex (or PM / Scl complex) is a nucleolar macromolecular complex with ribonuclease properties  . Thus, it is important in mRNA degradation and ribosomal RNA processing  . Exosome complexes are found in both eukaryotic cells and archaea.  . It is therefore incorrect to use the names exosomes; and exosome complex; interchangeably. As a result, it is necessary to follow guidelines of International Society for Extracellular Vesicles concerning proper nomenclature.

Conclusion and future perspective need to be extended. Specifically what are the advantages of such therapies include numerous factors such as prolonged drug half-time, maximizing biological compatibility, less systemic toxicity. So what is still blocking the implementation of exosomes into clinical trials that so little progress has been made on this issue. Please indicate based on analyzed publications what are the major limitations.

Author Response

Please include proper exosome classification according to International Society for Extracellular Vesicles. It is therefore incorrect to use the names "exosome" and "exosome complex" anld please indicate the difference.

I appreciate your work in the field of exosomes. However please add section on classification of exosomes to clarify that all publications that are included used the same type of vesicles.The discovery of exosomes was made byconfirming that the transferrin receptors in reticulocytes are associated with small vesicles, which are then secreted from the maturing reticulocytes into the extracellular space . The  term exosome; was clarified a few years later by Rose MamelakJohnstone, the head of the Department of Biochemistry at the Faculty of Medicine at McGill University in Canada 27 . In 1997, a structure called the “exosome complex” or - often wrongly - the human “exosome”, was additionally discovered. These two concepts should not be confused, as the exosome complex (or PM / Scl complex) is a nucleolar macromolecular complex with ribonucleaseproperties  . Thus, it is important in mRNA degradation and ribosomal RNA processing  .Exosome complexes are found in both eukaryotic cells and archaea.  . It is therefore incorrect to use the names exosomes; and exosome complex; interchangeably. As a result, it is necessary to follow guidelines of International Society for Extracellular Vesicles concerning proper nomenclature
Conclusion and future perspective need to be extended. Specifically what are the advantages of such therapies include numerous factors such as prolonged drug half-time, maximizing biological compatibility, less systemic toxicity. So what is still blocking the implementation of exosomes into clinical trials that so little progress has been made on this issue. Please indicate based on analyzed publications what are the major limitations

Authors' response: The authors appreciate the constructive feedback from the reviewer and we are thankful to the reviewer that because of reviewer comments our article got more improvement and we enhanced the quality of our article as compared to its initial draft. We have added tables as per your worthy suggestions. Please see the revised manuscript for confirmation and further suggestions. We have glad to inform you that we have updated the conclusion parts accordingly based on scientific data.

Reviewer 3 Report

·         The manuscript lacks character of in-depth review and mainly focused on general aspects of exosomes delivery, which is available in many reviews. The key challenges, stability and delivery concerns specific to central nervous system diseases are missing. The reader may not obtain a clear view due to introductory level of explanation. There are many reviews published on the topic exosomes, including exosomes delivery for central nervous system diseases. Authors should clarify the need of another review on this topic. Also, cite summary of few previously published reviews in the introduction section. The manuscript has only one table and the figures present in the manuscript was already present in many exosomes review articles.

Author Response

The manuscript lacks character of in-depth review and mainly focused on general aspects of exosomes delivery, which is available in many reviews. The key challenges, stability and delivery concerns specific to central nervous system diseases are missing. The reader may not obtain a clear view due to introductory level of explanation. There are many reviews published on the topic exosomes, including exosomes delivery for central nervous system diseases. Authors should clarify the need of another review on this topic. Also, cite summary of few previously published reviews in the introduction section. The manuscript has only one table and the figures present in the manuscript was already present in many exosomes review articles.

Authors response: The authors appreciate the constructive feedback from the reviewer. We have added tables as per your worthy suggestions in the introduction and conclusion parts. Please see the revised manuscript for confirmation and further recommendations.

Round 2

Reviewer 1 Report

The authors have addressed the issues raised by me.

Reviewer 3 Report

The revised manuscript still lacks in-depth review. Since there were many published papers on this topic, and considering the Journal standard, I would reject the manuscript in its current format.